# "Bad Students" and the Configuration of School Failure through Their Social Representations

Ángela María Velasco Beltrán * and Rocío del Pilar Velasco Beltrán

Faculty of Educational Sciences, University of Granada, 18012 Granada, Spain; pilyvedoc@correo.ugr.es
* Correspondence: anmavelbe@correo.ugr.es

**Abstract:** Social representations can influence, to a great extent, the way in which we relate to people in different situations. In the educational environment, these representations—in terms of their adaptation or not to the school's academic and behavioural demands—can lead to the school failure of those considered as "bad students". Following an assessment, interviews and discussion groups were conducted with various members of an educational institution in the south of Bogotá to describe the social representations that prevail in this community about students who were at possible risk of school failure and how these representations could determine whether or not the student does indeed fail. Discourse analysis results yielded categories such as the differential construction between a "good student" and a "bad student" as well as stigmatization and conflicts with the school hierarchy and also showed how, occasionally, student failure is considered a consequence of family or social and economic factors unrelated to the school itself and to pedagogy. In conclusion, it is evident that there is an urgent need to establish institutional mechanisms that promote and adopt inclusion in educational relationships and practices from the perspective of the needs and interests of the most vulnerable pupils.

**Keywords:** academic achievement; behaviour; evaluation; teacher–student relationship





## 1. Introduction

The school, as a social institution, establishes, legitimises, and validates basic learning and ways of being that allow students to act according to the established standards, thereby becoming guarantors of the social order and a reproduction mechanism of the current system (Vázquez 2018; Tarabini 2018). Satisfactory results in evaluations and appropriate adaptation to institutional norms determine (inside the school) some classifications, such as those that categorise "good" and "bad" students. The first group is considered competent, responsible and autonomous; on the other hand, students who present different behaviours from what the school expects, who do not pass the year or who do not "adapt" to the institution's established profile are designated as "bad students" at risk of school failure (Marchesi and Gil 2003; Susinos et al. 2014).

This conceptualisation is determined by the changes in the different educational systems that result from the historical moment, the culture, and the educational model in which an educational community is immersed. Being a "good" or a "bad" student depends greatly on the magnitude of difficulties that a pupil may encounter during their school years in adapting to the demands and evaluating gaze of each school (Marchesi and Gil 2003; Marchesi 2004). These situations are "the expression of how much, in educational terms, we still have to do" (Marchesi and Gil 2003, p. 14) and lead to school being a possible obstacle to equality and inclusion, rejecting difference and demonstrating the limitations of the school to accommodate it (Tenti 2008; Susinos et al. 2014).

Describing the social representations that prevail in educational communities about students who are possibly at risk of failing at school and how these representations determine whether or not this actually occurs allows us to understand how these situations are

configured and to identify the reality of the school and the knowledge that is produced, not only scientific knowledge but also everyday knowledge. Social or everyday knowledge is created in every social scenario—such as schools—through human interaction and includes symbolic, affective, and cognitive contents that in some way determine life, whether that be its forms of organisation or communication (Berger and Luckmann 1966).

Systems that are constructed recognise beliefs, stereotypes, opinions, classificatory logics, values, and norms, and they define a collective conscience that governs with normative force by establishing limits and possibilities in the way community members act. These systems are known as social representations, and they arise from the interconnected relations that are typical of interaction, in which each community member can accept, reject, or stigmatise other people through the construction of their own social representations.

Undertaking studies into the representation of a subject (like a good or bad student) leads to the recognition of the modes and processes that constitute social thought, through which people construct and are constructed by social reality. Therefore, it is essential to understand how social representations appear, how they produce knowledge from daily life and how they emerge in groups when discourse and communication suggest shared or divergent points of view and perceptions on various issues, situations, or people (Moscovici 1976; Jodelet 1984).

Some studies carried out on the subject of school failure from the perspective of social representations in Spain (Marchesi and Gil 2003; Aguado López et al. 2009; Vázquez 2018) examined parents, teachers and students' opinions on school failure. They then identified some internal causes (students' disinterest and low effort and a lack of support from teachers) and external causes (related to the family, the educational system and uninteresting contents and classes). These studies have theorised about the impact that these representations can have on students and their schooling process, suggesting that failure leads to mistreatment that ends up justifying exclusionary and discriminatory educational practices, which in the long run lead to early school leaving, poor working conditions and low income, ultimately resulting in trigger poverty and social injustice.

Other research carried out in Argentina by Federico Butti (2018) and María Paula Quiroga (2013) thematised the social representations that circulate concerning possible adolescent school trajectories in working class sectors of society in relation to school success or failure. Their ethnographic studies established the mediation of representations in social interactions and characterized the relationship between the students' social class, the timing of school hours (in Latin America, due to a lack of infrastructure, there are often two cohorts of students that study in the same school: one in the morning and one in the afternoon), the location of the school, teachers' expectations, and parents' educational background as possible determinants of adolescent school failure. Finally, the research conducted by Zamudio Elizalde et al. (2019) in Mexico used questionnaires to analyse, from a quantitative perspective, which elements of the social representations of high school students have an impact on school failure. All this research has been taken as a key input in the development of this analysis.

Therefore, this study represents the possibility to make an analysis of a school in Colombia, from the perspectives of different stakeholders and through their own voices and feelings, in order to answer the following: What are the processes that put "problematic students" at risk of school failure in an official educational institution in the city of Bogota, and how do they come about? What are the social representations that teachers, parents, and classmates have about these students? What type of institutional responses are there for this type of student?

In this way, we will describe the social representations that predominate of students at possible risk of school failure in one educational community and determine if and how these representations determine the actual risk of failure occurring, as it is a phenomenon that continuously develops inside the school and can be approached from different perspectives within national, local and institutional environments.

### 1.1. Literature Review

1.1.1. School Failure: A Multicausal Reality

School failure is a term that, at a general level, not only designates students who do not complete compulsory education (Calero et al. 2010), but also denominates, measures or quantifies a negatively charged reality by explaining it with "apparently depoliticised" figures, which end up being individualised within the school (Vázquez 2018; Tarabini 2018). Through various studies, it has been found that school failure is synonymous with students' disengagement with their learning (Marchesi and Gil 2003; Escudero et al. 2009), with the school (González 2017) and with their own grades (Vallejo García 2018).

Hargreaves (2003) establishes differences between what he calls schools that fail and failure inside the school. For the first situation, he considers that it is necessary to carry out institutional change, which includes all members of the school community—principals, teachers, parents, and students—all of whom must recognise the situation and specific actions for improvement. As for failure inside the school, he considers it as a possible form of exclusion based on the idea of capabilities and individual conditions that separate and divide people: "those who have not succeeded, those who fail, become victims of distinction, the object of the displeasure and scorn of others" (Hargreaves 2003, p. 237). In this way, a school that fails is considered an institutional problem, while failure inside school is an individual problem.

For the OECD (2012), school failure is an unfortunate consequence of inequity and exclusion due to the lack of opportunities, the most obvious example of which is school dropout. School failure is a term used in various realities: pupils with discordant behaviours, those with low academic achievement or those who skip or have dropped out of school, among others. It also assumes all failures are the same, building a subjectivity and image of these students as "failed subjects" who fail to make progress in different life areas. This assumption negatively affects students' confidence and self-esteem, focusing the problem on them and forgetting the responsibility of other actors such as the family, the state or the school (Vázquez 2018). In other words, school failure is the result of an accumulation of misunderstandings, tensions, struggles and unfortunate experiences in various contexts, and something that requires us to look at the educational system and its relationships, rather than just at students (Vázquez 2018; Escudero et al. 2009).

For this research, school failure is taken as a multicausal phenomenon and situation (Marchesi and Gil 2003; Escudero et al. 2009; Susinos et al. 2014; Vázquez 2018), which happens to some students who, once the school year or their time at school has finished, have not acquired the minimum knowledge, skills or abilities needed to pass the year. These students do not "adapt" to the standards, and their undesirable or unexpected behaviour does not fit within the boundaries set by the educational community (Romo Torres 2018).

In search of a referential framework that provides access to the school space to understand the way in which this situation is configured from social knowledge that is built from and about the subjects that inhabit educational scenarios, social representations are presented as spaces of recognition that may contain a variety of situations such as school failure that, when crystallised through a word, a gesture or an encounter, allow for a much closer reading of any reality.

1.1.2. Social Representations and the Construction of the School Reality

Representations define a common frame of reference that allows social exchange, transmission, and dissemination of "spontaneous" and "innocent" knowledge. Representations make it possible to understand and explain reality, to acquire knowledge and to integrate it into a framework that is comprehensible to the individual (Moscovici 1976). Moreover, representations bring us closer to the "world view" that people or groups have since this knowledge is used to act or take a position before other subjects based on language and communication that construct and give meaning and sense to reality (Berger and Luckmann 1966). Addressing social representations makes it possible to understand the dynamics and the determinants of social interactions since representation, discourse and practice are

mutually generated (Rubira-García et al. 2018; Moscovici 2019; Gutiérrez-Vidrio 2019). This all underlines the importance of knowing, discovering, and debating their origin, from which a representation and therefore a social practice can be modified.

Moscovici (1976) defines social representations as a system of values, ideas, and practices that, on the one hand, establishes the order with which individuals orient themselves in the social and material world and, on the other hand, enables communication by providing a code with which they classify the different aspects of reality and their individual and group history. Social representation is a construct that arises from the perception of and previous knowledge regarding a subject (Howarth 2004).

In the same sense, Jodelet states that a social representation is socially elaborated and shared knowledge that helps to understand reality and give it a meaning and a sense; it is knowledge with a practical character in daily life, as it gives possibilities to accept or reject the other person, to know how to act in front of them and to assign them a place in society.

> *Social representations concern the knowledge of common sense, which are available to be used in daily life experience; they are perceptions, constructions with the status of a naive theory, which serve as a guide for action and an instrument for reading reality, and systems of meanings that allow us to interpret the trajectory of events and social relations; they express the relationship that individuals and groups maintain with the world and other people. These representations are forged in interaction and contact with the discourses that circulate in the public space; they are inscribed in language and practices, and they work as a language because of their symbolic function and the frameworks they provide to codify and categorize what makes up the universe of life.* (Jodelet 2000, p. 10)

This approach combines cognitive and social dimensions oriented towards the communication and understanding of the social environment and reality as a process to create this "common sense" knowledge. Determining how a community member constructs a representation of a pupil in a possible school failure situation implies knowing the dynamics and factors that influence this construction through community perceptions and experiences. Thus, a social representation is consolidated as the interpretation made by a community about a studen, from dimensions such as information, attitude, and the field of the representation that is built (Moscovici 1976, 2019).

As it is not possible to separate a representation from the everyday practices that are inherent to it, when we study representations, we first look at the contents and processes of social learning about the object of the representation as a collective construction. Representations give a deeper insight into the knowledge mechanisms to understand what is at the basis of human actions, they facilitate perception and action in the face of an external variable or unknown agent, and they become a scheme that gives order to the space that surrounds us and gives meaning to facts (Moscovici 1976).

Elements of representation such as a norm, a stereotype or an attitude are determined by the frequency of their appearance in the discourse or even by the number of links they establish with other elements. Other elements that speak of experience and depend on the context circulate are hierarchised and contain selected and interpreted information, value judgements, stereotypes, and beliefs (Abric 2001). This article evidences the portrayal of students in a situation of school failure (who are considered bad students) in the discourse of a specific educational community and the way in which other discourses are positioned around them, discourses that show the desire for school success but also individualise failure and generate situations of stigmatisation and rejection.

## 2. Materials and Methods

This study was carried out in a state school in southern Bogotá that offers both primary and secondary (lower and upper) education to a low-income and highly vulnerable population. This institution produces low results in national standardised tests (tests that measure educational quality annually) and has high failure and dropout rates (Secretaría de Educación del Distrito 2021). The research team belongs to this institu-

tion; it is composed of the director of studies and educational counsellor. We designed a qualitative study with an ethnographic approach to give a voice to the students and to the educational community in general, giving meaning to their interactions and sense to their representations (Bolívar et al. 2001).

This study approached the educational space from the sociology of discourse and from an ethnographic perspective to enrich theoretical development through the use of interviews and work in discussion groups.

The stories of students and other members of the educational community allowed us to understand the richness of their experiences from their imaginaries, disagreements, desires, and interests (Connelly and Clandinin 1995). The study subjects were not selected based on psychological or demographic characteristics; rather, the initial criterion used was if the subject was a part of the researched group (Gutiérrez-Vidrio 2019). Three students were selected as participants: two boys (12 and 16 years old) and one girl (14 years old) who study during the afternoon (the school teaches another cohort in the morning, and in the imaginary of teachers, the afternoon group has a greater number of conflictive students) and who belong to the 6th and 7th grades in secondary school. These students are considered by teachers and peers as students at risk of possible school failure given their characteristics of low academic performance (low grades, frequent letters to guardians due to difficulties with academic commitments, lack of interest and other issues) and disruptive behaviour in relation to institutional norms (frequent tellings-off, being monitored by the coordinators, incompliance with commitments made following misconduct related to school convivence and notes in their student record). The 6th and 7th grades were selected for this study because they are the grades with the highest failure rate among students in Bogota (Secretaría de Educación del Distrito 2021; Romo Torres 2018).

The discussion groups and semistructured interviews were conducted (between March and July) by the teaching research team, of which one author of this article is a member and which has been recognized by the educational community for work carried out over more than a decade. The interviews with students and parents were conducted at the counsellor's office, as it is a real space of trust, which is different from the classroom and which is conducive to calm and open dialogue (Table 1). Additionally, discussion groups were held with classmates of the subjects and their teachers. Two discussion groups were held, one with 14 students from the class to which the teenagers that are the focus of this research belong and one with 5 teachers who were in direct and weekly contact with these students through their classes and who expressed interest in participating in this research. These instruments were alternated weekly to not saturate the community with activities different from their daily work, and the schedule was flexible to ensure that they took place at a suitable moment for the participants.

**Table 1.** Techniques and instruments used.

| | Technique | Instrument | Data |
|---|---|---|---|
| Each Student | *Semistructured interviews (5):* Made by the teaching research team. 2 interviews with each student (45 min) 2 interviews with each teacher (45 min) 1 interview with the student's parents (1 h) | Interview script | Interview transcribed from audio recording. |
| | *Discussion groups (2):* Run by the teaching research team. With peers, without the presence of the student in question (40 min) and with 14 students With 5 teachers (40 min) | Discussion group script | Discussion group transcribed based on audio recordings and some observations made during discussion groups |

Source: Made by the authors.



The names of the participants were anonymised to comply with international data protection regulations and the Granada University research code of ethics. All meetings and interviews were carried out with the informed consent of the institution, the parents, and the students themselves (Table 2). The interviews were structured using open questions, which allowed for a greater expression of experiences. After each meeting, the teaching research team transcribed and showed the manuscript to the participants so that they could express their agreement or disagreement with the way their ideas and feelings were captured in it.

**Table 2.** Scripts of the instruments used.

| | | Questions | |
|---|---|---|---|
| | | *Principal* | *Secondary* |
| Semistructured interview script | Student | What do you think your family, teachers and classmates think about you? | How are you doing at school, and how do feel here? <br> What kind of difficulties do you have inside the school? <br> Why does it happen? <br> What is your relationship like with your teachers and classmates? <br> What kind of repercussions has this relationship had for you? <br> What do you think about what happens in the school? <br> What kind of sanctions or actions has the school administeredregarding your behaviour? <br> How have you felt in those cases? |
| | Teacher | What do you think about students who present difficulties (academic or with their behaviour)? | At what point do you consider a student is having difficulty? <br> How do you describe this kind of student? How do you interact with them? How do they interact among themselves? What are their learning processes, learning rhythm and behaviour? <br> What happens to these students? <br> What are the reasons for their difficulties? <br> How do these difficulties affect your classes? <br> What consequences has this situation had on the student and the whole group? <br> What are the institutional guidelines to manage these cases? <br> What is the process you follow to manage a student who presents difficulties (academic or behavioural)? <br> What can you do to help this student? <br> What would be the best way to act in these cases? |
| | Parents | How do you think about your child as a student? | How do you think your child feels at school? <br> How is your child doing at school? <br> What are your child's strengths and weaknesses? <br> How has your child's school life been? <br> Why do you think they behave in this way? <br> What consequences has it had for your child? <br> How do you think you can give help to your child? |
| Group discussion script | Student's Group | | In your opinion what kind of students should be at school? <br> What are the main difficulties your fellow students have? <br> What are the reasons for these difficulties? <br> How do we act and how should we act to face them? <br> How can you determine that a teenager has an academic or behaviour difficulty? <br> In which cases should a student be excluded? |
| | Teachers | | Are there any children who have difficulties in the group? <br> What characteristics do the students with difficulties have? <br> What kind of difficulties do they have? Why do you think they have difficulties? <br> How do these students behave? What is your opinion about them? <br> What are these children's relationships like with their teachers? How is your relationship with them? <br> What does the school do to understand students with difficulties? <br> What do you think the school and teachers should do in these cases? <br> What could you do about this situation? |

Source: Made by the authors.

Once the interviews and discussion groups had been conducted, they were transcribed. Each document was identified according to its origin (T: Teacher 1, 2 or 3; St: student 1, 2 or 3; P/T: parent or tutor; Int: Interview 1 or 2; DG: discussion group) and paragraphs were numbered (P1, P2, . . .), for example, St1.Int2.P6. After this, emerging categories and subcategories were identified, and key paragraphs were organized in tables by category, and thus the information obtained from each research member (Student, Teacher, Parent) was summarised.

A discourse analysis of the themselvesnterviews and discussion groups was carried out by the same research team. In both cases, the large amount of information needed to be broken down into groups of features and categories to find their sense and meaning, considering that this type of analysis allows for an objective, systematic and quantitative description of the content manifested in the discourse. Social representations, more than just organising ideas, are a sociocultural construction that arises from situations of exchange and in relation to the conditions under which the discourse occurs. They are influenced by the actions of the student in their group, and they arise in a context of relationships and communication that transcends the internal reality of the subject (Gutiérrez-Vidrio 2019). Therefore, it is necessary to reconstruct them to know who produces the discourse, where and in which situations they do it and to whom it is directed.

Following this, we proceeded to organise and categorise the information manually to highlight the ideas in the speech and quantify them depending on the meaning or topic which they referred to. The discourses were compared, and the information was hierarchised, with the intention of determining the main representations and their structure. Thus, the data were grouped into homogeneous sets that allowed us to identify relationships and inferences between the various topics analysed. Finally, the report was written.

## 3. Results

To analyse the results, it was necessary to understand how diverse social representations emerge in educational settings and how they configure a process of possible school failure from the differential construction between what represents a "good student" and a "bad student" via stigmatisation and confrontations with the school organisation to the increased risk of failing the school year, accompanied in many cases by exclusion.

### 3.1. The Construction of the "Good Student"–"Bad Student" Dichotomy: Desires, Hopes and Realities

All students arrive at a school with their history, expectations and interests, and they must adapt to institutional and group dynamics, a situation that may or may not be easy since their desires or interests are not a criterion when selecting an institution from those on offer.

In the institution, teachers follow the established school curriculum to determine skills, knowledge and behaviours that they consider a "good" student should demonstrate according to the student profile (in the case of the institution where the research was conducted, this profile describes a student that participates in teamwork, with quality in the education process, and who builds a life project based on a good axiological foundations). Some differences are established between "good" behaviour and "inadequate" behaviour, and it is determined what is allowed from students and what is not.

Thus, some students who are seen by their teachers and classmates as students who do not "adapt", who frequently break rules, who confront the teachers and who seek the attention of the class in order to make themselves visible begin to be considered "bad students" because they do not match the institutional student profile: they are "unpunctual" with class start times or in delivering assigned activities, they confront or challenge teachers, they have several notes in their student record because of their constant disrespect or irresponsibility, or they are always distracted; they "show little interest and indifference", "they hinder the work of others" or do something different while other students work, or they stand out because they "avoid classes", "are lazy", "do not study" and "do not do the minimum to pass".

The school, represented by its teachers, establishes differences between students and carries out actions to make them behave as expected and become "normal" students. Thus, teachers can "give them opportunities" by explaining how to behave and how this can have a positive impact on their lives—academically, socially and familywise—so students become people who "are clear about their goals in life [...], who know what it means to study and what the school setting implies" (DG2.P19).

These students are expected to immediately fall into line, and much more is demanded of them. If they make a mistake, their negative image is reinforced, and it is shown that they are not suitable for school and that they are "bad students". Unfortunately, in these cases, teachers take the conflicts personally and seek to impose their authority: leaving them out of the classroom, excluding them from activities, giving them a negative image of themselves and warning them that "they will make a note in the student record, mum or dad will be called to the school, and they will have to sit out classes until their parents or guardians respond to the institution's call".

> *Here the sanctions are to sit for one or two days in the coordinator's office [...] No, they send them home this year (smile) [...] when they should be given social work or something that really, I mean, leads them to reflect. I mean, here, first we must look at the type of sanctions that are being imposed or sanctions that the people in charge of doing it are imposing [...].* (T3. Int1.P17)

> *Sending them to the school counsellor and all of this? (Derogatory tone) [...] In the case of...7th graders, last year, I can't give you any insight because I was not involved in their case. The student's case, the one I was most involved in, they were talked to...to them and his parents even took him to the EPS (Health Service Provider Company) to see a psychologist [...] I must have the little sheet where the specialist gave some recommendations.* (T1. Int1.P19)

These situations are rarely perceived by teachers as situations that concern them as well. They generally assume the situation to be the student's problem to be solved at home or through the school counsellor. In their opinion, the school does not have sufficient tools to manage this problem internally and for this reason, they resort to external authorities. Teachers' solutions are more oriented towards sanctioning or being indifferent.

> *Well... it's not great... (silence); as I told you, there are people who keep their eye on us, and many teachers judge us without knowing, and it is horrible to hear how they talk about us: here comes the gang of arms and drug traffickers! You feel bad, besides it's all gossip; the teachers take a position based on what others say and based on appearances. They do not know the people well, they assume that everything is the way they are told, and that's where the problems start, the teachers listen and spread it: this one tells the other one, and the other one tells the other one, and so it goes, and the gossip goes on and on and on. That's when you get annoyed with some teachers, and you trust in others.* (St1.Int1-P11)

Teachers attend to their students' difficulties in different ways: some within what is established in the Student Conduct Handbook (also known as the "manual") and others on a personal basis according to what they believe should be done. Some teachers refuse to consider the institutional process in the Student Conduct Handbook, and they delegitimise it to regain control and power to prevent chaos from being generated: to bring "justice" and prevent "bad students" from continuing to "mistreat" the community. Other teachers are simply indifferent and almost ignore "bad students".

> *What happens is that the manual has a problem, like the Constitution: it says many things "in words". But in practice, when it is time to apply it, it does not happen because the school itself is not legitimate in its actions [...] so that moment is when one loses the credibility to apply the manual; when those who should apply it do not use it, then these conditions are generated [...]. He was very haughty with me twice: no problem! I almost had to apologize to him. So that doesn't work here: this is why students generate their own ways of relating to each other: because they know that the manual is no longer valid [...] It is very general; that is, there should be a protocol for specific situations, and there is not.* (T1. Int1.P4)



*3.2. The Stigmatising Dimension of School Failure: Prophecy and Destiny*

The majority of state schools in Bogota serve a population with low socioeconomic and cultural status—with unsatisfied basic needs and a low level of family schooling. A good part of the social representations that are constructed around students at risk of school failure also come from the place where they have grown up and/or where they live:

> *How is your neighbourhood? Normal (silence) What do you mean?... Well, normal: there are robberies, mugging;, but since they already know us, they don't do anything to us. There are good people and bad people as in any place and in every neighbourhood; there are people who feel superior. In the neighbourhood, there are different areas: the park, "la olla" (literally "the pot", used for a rundown zone where drug addicts reside) and the block of "recyclers" and bars.*

> *How do you feel in your neighbourhood? I feel good, I know it and they know me, and I feel safe because I am from the neighbourhood. There they identify who is from the neighbourhood and who is not. It's not as unsafe as they say. Strangers see us in a strange way, they judge us, they see us as less [than them]. (St1. Int1.P2)*

Influenced by this, many teachers have low expectations for their students, possibly on the assumption that families or social contexts with difficulties entail disadvantages in academic performance. On many occasions, the family is judged for not doing its part for the child's education, shifting the explanation of success or failure to individual situations.

> *As you can see, yes... yes, because here there are children with different... as I said, with different types of education because of their family, their socioeconomic level, their beliefs; because of many things, there is a huge variety of children [...]. Other children, depending on the level of support from their family, well, one works with them. There directly with Instituto Colombiano de Bienestar Familiar [social services] located in the Timiza neighbourhood, last year, curiously, one person even helped me, through the Family Commissariat, and curiously he helped me with a psychologist he had there. So, it made it much easier for me to handle the cases because I worked on a case directly there, and it worked very well, but because of that help, right? And other cases it's like... There are some that, how should I say—they are like on "Stand By", or students leave, or for example, a case right now [of a pupil] in 703. It is a problem between the mother and the stepfather, and they are going to take away him and his sister too. (T1. Int1.P3)*

Another aspect evidenced during this research was the "hidden" way in which these representations created about the students are constructed and transmitted, which leads to these representations not being expressing openly; rather, they are circulated in halls or in informal chats and not in institutional spaces such as teachers' meetings. They are created in informality but end up occupying institutionalism effectively, and they end up being accepted to avoid interpersonal conflicts because they come from figures with greater recognition in the community, who endorse and justify them.

> *For example, to Professor "T", because he said that we were selling, that we were arms and drug dealers. If a student is permanently told that he is something [like that], they make him make bad decisions. (St1.Int1. P11)*

Thus, when a "negative" social representation is constructed of a student, the views focus on the faults and mistakes they may make; meanwhile, their abilities are ignored, the level demanded of them is increased, and it is expected that the student will change and transform their behaviour following a telling off. When these expected responses are not obtained, this leads to prejudice and a sense of unease.

*3.3. The Attitudinal Dimension of School Failure: "This School Is Not for You"*

School difficulties affect the academic environment, which contradictorily values behaviour more than acquiring a certain level of knowledge in the subject. Even though the teaching staff may recognize this, they give failing grades to students with behavioural problems:

*[He failed the subject], but more for irresponsibility because he did not complete the tasks, but in terms of ability, he has it because there are other children that we as teachers, we see it is difficult for them, that they could not do the tasks [. . .] No, not at all, with me, he works, he does the things that I . . ., well let's say that. . . at the beginning, I asked him for things, but he was kind of lazy, so I know, why do I make the effort to write [in his notebook, or the student record]. . . I marked [what I received from him]; that's why he failed. There were days he was absent; he never showed up to tell me: teacher. . ., so I added up the grades he had, and he failed. (GD1.P3)*

Despite the low academic performance that characterises many of these students, they are recognised as frequent attendants of the institution, possibly motivated by maintaining a social and emotional link with their classmates:

*And what does he like about school? (Silence) I think he feels bored at home; well, the. . . the girls—he has a lot of female friends. I tell him "Oooh!" because he is very sociable, because his friends begin to come to our home: (Hmmm) one girl, another one, another one [. . .] He has a lot of friends, I mean, men and [. . .] girls and men, and well, they love him [. . .]! When it is his birthday and so on, they give him cards, they write him things. I mean, he is very sociable; well, I think he misses it at home, and he likes coming to school [. . .] Well, you see that his classmates go to look for him, and they are from here at school because you see them in uniform: they are all from the same school, right? They go. . . and sometimes one afternoon they watch movies, and you see that they are good kids! (P/T1.Int1.P3)*

*Why do you feel good even though you are not good academically at school and teachers tell you off? Because I spend time with my friends and play with them; I laugh. (St2.Int2.P4)*

However, despite their qualities as people, they come into conflict with the institutional norms that some teachers legitimise and validate in their classes or the demands that individual teachers make of students.

*They get used to the norms that apply in the school [. . .]. In that way, these norms must be established so things work [. . .] And what happens is that the hidden curriculum that the students handle plays an important role: knowing with whom they behave and in what way. In that case, we ourselves are to blame because they go to a teacher's class and in this teacher's class there is a certain behaviour, and if they change teacher, there is another behaviour; so here, as teachers, we are all heterogeneous; there are no rules that make us all behave in the same way. So the children, in their hidden curriculum, say "let's go to such and such a class, and they know we go to the bathroom, we go to the playground, we go to buy a soda, and then if we go up to the classroom, we do not have any problem". So, I think that, as such, the children do get used to the rules of the institution; it is clear, they get used to norms [. . .] Or, at least, to the rules that each teacher has". (GD2.P21)*

The rejection of these students and the inadequate or indifferent handling of their situations increases the problems in which they are involved. Stigmatization, the use of disparaging language, prejudices and the harshness with which they are told off are some circumstances that damage their self-esteem, increase their resentment, affect their interpersonal relationships (peers and parents), demotivate them in their learning and end up reinforcing behaviours that are criticised, placing them in situations of greater risk.

*Well, if for whatever reason, you don't do homework, then they start yelling at you and tell you off for everything, so I behave badly. (St2.Int1. P3)*

When the expected changes in the attitude of this student do not occur, the student becomes more "rebellious": they tire of staying quiet about the negative comments they constantly receive, and they react. This act is taken as a sign of disrespect, their parents are summoned, and some teachers begin the job of tarnishing the student's image in front of them, so parents feel their child is getting worse and worse and believe the school is not the best place to be. This generates conditions such that in the next conflict, the guardian (parent) gets tired of the situation and decides to withdraw the student.

When this happens, some teachers feel proud because they "solved" the problem and demonstrated who had more power and was right; they reaffirm their authority through fear and consolidate the veiled idea about the good behaviour with certain teachers because they can "get them out of the institution".

> *Well, firstly we should not bethemselvese: last year, people who were causing a lot of trouble were taken out of the school: those who scammed, those who extorted, the violent ones, the hooligans. So, that had an impact. Secondly because, as the "prize issue" [avoiding punishing students with poor behaviour] had gone too far and the boys had started to understand that the prize issue came from the headteacher's office, then we had to generate our own ways to solve the situations: relocation or expulsion.* (T3. Int1.P5)

The lack of institutional organization, the ambiguous handling of the situations and the disregard for rights and of the Convivence Handbook by a great number of members of the community are some of the situations that end up institutionally justifying the withdrawal of students with characteristics that they consider incompatible with how they should behave.

## 4. Discussion

At school, behaviours and abilities are naturalised and normalised, and based on them, categories of students are constructed: those who are intelligent, those who are responsible, those who are lazy, those who never achieve and those who would seem not to want to achieve (Tarabini 2018). These classifications are not natural, but they have their origin in school, social and family relationships (Romero 2021). Additionally, these labels and other situations affect the way students are seen by others and undoubtedly in their school life since representations are built about some of them that at first, classify them as "bad students" and later, if behaviours discordant with the established norms are added, label them as problematic students at risk of failing and even being excluded (Alcaraz and Gómez 2014). "The essence of a representation is to make people understand and share an idea with the same vividness as a perception or an emotion, and vice versa" (Moscovici 2019, p. 15).

A student's life conditions determine, to a great extent, the way they hey construct themself as a subject and relate to other people. Some typical life conditions in Bogotá include, for example, if the student comes from a family where the student receives care, if they have a father or mother who has received education, if they have a large family or one with economic difficulties and if they arrive to school hungry or with other needs (Romero 2021). It is important to emphasize that family economic level plays a decisive role in teachers' expectations, and when they do not know a student's situation, they assume it is negative and speak about it in a pejorative and prejudiced way, referring to "unfortunate or unstructured families" and ignoring societal problems (Tarabini 2018).

Many teachers' expectations are determined by this economic status, which is associated with success or failure, with greater or lesser intelligence or with the effort made in activities, tests, or exams, among others (Romero 2021). The well-known "Pygmalion effect", in which expectations are a destiny to be fulfilled (Kaplan 2008b), can generate real effects on students, as "education is a deeply emotional practice and, through their expectations, teachers transmit positive or negative emotions, attributes, stereotypes, stigmas, affections and disaffections, which characterise educational relationships in the everyday" (Tarabini 2018, p. 61).

Classmates or the class group also play a decisive role as the other significant subjects who inhabit the school (Tarabini 2018); in other words, they are people who determine the self-image and the world around students and who influence their actions and relationships. Additionally, the social representations that are built of students at risk of school failure are also influenced by the representation of the place where the educational centre is located or which its students come from (Quiroga 2013). In this case, as the school is located in the south of the city, which is an unsafe residential and commercial zone, the location arouses fear and some repulsion towards its inhabitants since frequent criminal episodes occur there (armed robberies, physical assaults and microtrafficking, among others).



The representations that teachers construct of their students determine some of their practices (Butti 2018), especially in the case of "bad students": tellings-off, shouting, observations in the student record, summons to parents, suspension, and expulsion from the school, among others (Howarth 2004). In this sense, representations take shape in other areas of the school as well:

> *"The teachers' room" [...] is a space for interaction among teachers, where many assessments, both positive and negative, are transmitted about different students. The classroom is one of the spaces in the school where the differentiation process takes place, which is reflected in preconceptions about better and worse students.* (Alzás and Pelícano 2018, p. 174)

The common denominator of the response to situations involving these students is to exclude them from the classroom, from activities or from the institution itself. Moscovici (1996) analyses what happens in this sense, when he states that social influence aims to maintain and reinforce social control as, for it to exist, it is necessary for the educational community to have the same values, norms and judgement criteria (both inside and outside the school), which exclude those who refuse to change. Those who are considered "bad students" or "troublemakers" are led to believe that they are wrong, that their behaviour is abnormal. The group in power—the teaching staff—seeks uniformity (eliminating particularities and individualities) and employs cohesion and attraction as strategies to attract new members and reduce the distance with dissident minorities (Moscovici 1996).

Social representations materialise in cultural agreements, such as acting in one way or another in certain scenarios such as school (De Alba 2022) and rejecting strange behaviour when it deviates too much from the norm, as it threatens the established order.

The representations teachers construct around "bad students" or those that are at risk of school failure are constituted in relation to what they consider a "good student" or a "successful student" and transmit perceptions of reality that influence the school environment (Tarabini 2018; Vázquez 2018). The judgements that are made about someone depend, to a large extent, more on the social representations in which they are included than on their intellectual capacities and behaviour (Moscovici 2019). These representations are explicit both in discourse and in practice, and they acquire further legitimacy due to the authority given to teachers. They seek a uniform student group largely because they were trained under beliefs and norms that demand a nonexistent homogeneity and which consider that it is the student who should be standardised and adapt to the school (Tarabini 2018). The educational system's demand to value students according to their knowledge, skills and behaviours and the absence of teacher training to manage diversity have generated pedagogical practices that must be eradicated from the school.

## 5. Conclusions

School failure is one of the main problems that must be tackled using diverse educational policies, and in this way, it is imperative to assume that "educational centres, far from being mere large-scale social viewers, are active agents in producing processes of success, failure and school dropout" (Tarabini 2018, p. 73). Current educational policies based on standards prioritise results and do not tackle educational practices that discriminate, undermine teacher–student relationships and use standardised curricula and exclusionary methodologies that are insensitive to the realities in the school (Rujas 2017).

It is necessary to ask ourselves about school failure itself and its possible causes, as educational policies are interrelated with how this phenomenon is managed inside an institution and even more in terms of teachers' responsibility regarding failure. There seems to be a three-way solution: designing inclusive educational policies, nurturing a culture of inclusion and adopting inclusive educational practices (Vázquez 2018; Tarabini 2018).

Inclusive education policies should be based on teacher training that favours pedagogical relationships based on otherness and participation, which arouse the desire to know and which facilitates students' affective development. This training should aim to

help teachers as a whole change their expectations regarding those they consider "bad students", set high goals for them and allow them to generate confidence in their abilities (Marchesi 2004).

The school's responsibilities are to seek consolidation of a socially responsible teaching culture (Tarabini 2018): one that recognises the effects of the teaching–learning process based on empathy, that reflects on the possible biases of its practice and that is able to understand what is happening rather than just validating its supposed neutrality. This culture should be based on practices of respect, care, listening and otherness rather than prejudice, stigmatization, or authoritarian and exclusionary practices. In this culture, teachers, in accepting their role, understand that teaching requires more than just knowledge of their discipline: it needs a knowledge and pedagogy that are more empathic and a response to their calling as educators in the most general sense.

> *In adverse contexts, teachers in particular carry the social responsibility of alleviating and empathising with the social suffering of students [. . .]. The instances of reflection on practice can allow teachers to learn to know their students in terms of their identities, and material and cultural constructions, without prejudging them, without condemning them in advance; and as a result, they can be in better pedagogical conditions to interact with them.* (Kaplan 2008a, p. 13)

Although this research has some limitations in its implementation (perhaps in studying only a small sample of students from the same school), far from seeking generalisations, were able to identify the connection between the social representations of those who are considered "bad" students with their low academic results and the corresponding school failure. It is important to give voice to these children who are colloquially referred to as "bad students" and considered by the school as "problem cases" and to be receptive to what they think, what they feel, what they need from the school and the causes of their behaviour. As well as being a possible direction for future research with a narrative approach, the diversity and heterogeneity of students can and should be an area to be reviewed and recognised in teacher training

Understanding and getting closer to their students' world to deconstruct representations and stereotypes that exclude or discriminate is an unresolved task for teachers. It is therefore necessary to rebuild teacher–student relationships based on the recognition of the other and their abilities through dialogue to establish bridges of communication, affection and collaboration to manage diversity in the classroom. Today, the important thing is to connect with the lives of the students, their interests and needs and to engage them with learning and the institution so that they feel like an important part of it. Only in this way can we change the foundations of the current education system and make school a more welcoming place for students and their learning.

**Author Contributions:** Conceptualization, Á.M.V.B. and R.d.P.V.B.; methodology, Á.M.V.B.; validation, Á.M.V.B.; formal analysis, Á.M.V.B.; investigation, Á.M.V.B. and R.d.P.V.B.; resources, Á.M.V.B. and R.d.P.V.B.; data curation, Á.M.V.B. and R.d.P.V.B.; writing—original draft preparation, Á.M.V.B. and R.d.P.V.B.; writing—review and editing, Á.M.V.B. and R.d.P.V.B.; visualization, R.d.P.V.B.; supervision, Á.M.V.B.; project administration, R.d.P.V.B.; funding acquisition, Á.M.V.B. All authors have read and agreed to the published version of the manuscript.

**Funding:** This research received no external funding.

**Institutional Review Board Statement:** The study was conducted in accordance with to comply with international data protection regulations and the Granada University research ethical code.

**Informed Consent Statement:** Informed consent was obtained from all subjects involved in the study.

**Data Availability Statement:** Data from this research and support for the findings of the report can be found at the following link: https://internoredpedu-my.sharepoint.com/:f:/g/personal/avelasc4_educacionbogota_edu_co/EomyV9fCsMpJrT0TAQPhp0IBOWfQs7uMbGxI09Mjr-H61w?e=Q8B8vB (accessed on 22 July 2023).

**Conflicts of Interest:** The authors declare no conflict of interest.

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
