# Peer review of "“Bad Students” and the Configuration of School Failure through Their Social Representations"

_socsci, doi:10.3390/socsci12090514_

Round 1
Reviewer 1 Report
The manuscript analyses social representations of school failure in an educational centre in Bogotá. It is clear, relevant, well-presented and organised, and the research design is appropriate to answer the research questions. It presents consistent discussions and conclusions.
I would ask the author to specify in more detail the research instruments used, for example, the questions included in the interview or focus group scripts or, at least, the basic content around which they revolved.
The quality of English is good
Reviewer 2 Report
The paper addresses a topic of interest and uses an appropriate methodology to achieve the proposed objectives.
The results are clearly presented.
The conclusions are consistent with the results presented.
I think it is a good paper. But it is necessary to make some modifications that I summarize below:
1. Introduction.
There are some concepts that are repeated several times. For example, "common sense". It appears in lines 137, 140, 160, 172,..."
It is suggested to review the introduction to avoid these repetitions.
In general, the expression should be more direct, less convoluted, to better fit the style of writing scientific articles.
There are some verbatim quotes that are too long. Example: Lines 160-167, 520-524, 585-590.
It is suggested to leave only one or two of these quotes (in case they are considered essential) and use the indirect quote style in the rest.
In this way, the writing will adjust more to the style of scientific articles.
2. Out of date references.
In total there are 31 references.
Only 9 of them have been published in the last 5 years.
Although the topic of social representations is a "classic" topic, linked to authors from several decades ago, it is recommended to carry out new searches to locate recent studies related to this concept.
3. Limitations and future avenues of research.
It is suggested to mention some of the possible limitations of the work (very small sample, only one school is studied, etc.)
And possible future avenues of research on the subject.
It is also recommended to explain some of the possible practical implications of the results (for example in the field of teacher training).
4. It is necessary to review the text from a linguistic point of view.
There are several expressions that are copies of Spanish and that in English have a different structure.
5. Formal aspects.
There are some typos related to the use of "-". For example: "research, line 199"; "educational, line 200"; "richness, line 207".
There are also some typos in the spaces between words.
There are several expressions that are copies of Spanish and that in English have a different structure.
Author Response
Consulte el archivo adjunto

Reviewer 3 Report
The starting point of the article is very interesting. But unfortunately the conceptualization is confusing. Initially, I would have liked more problematization about what is meant by "bad students" and how this concept is connected to "school failures". Are "bad students" and "good students" the right words to use here? Maybe there is something that becomes difficult in the translation from Spanish? A suggestion is to consistently use "failed" and "successed" instead of "bad" and "good"? The fact that all the references are written in Spanish (?), there are none in English in any case, is an indication that the article fulfills a purpose of reaching out with important results and thoughts even to those like myself who do not master the Spanish language. As a reviewer, however, it becomes difficult for me to assess the relevance of the sources. This is especially clear in quotes, for example the one from Denise Jodelet (not Denis as in the references) where I wonder who translated the quote. Is it a book? Is it translated into English? (if use that translation!) Is there a French original?
The method is only vaguely described. I lack information in table 1 about how many from each category were interviewed, how many and who participated in the various discussion groups (or are they focus groups, both concepts are used, but it's not quite the same thing). What questions were answered? Used the term "bad students". How was the analysis done? Without knowing this, I unfortunately cannot comment on the results either, here you need to be much clearer. I would also like to problematize the fact that only one school and three students were selected, as well as the selection of these three students and how the fact that the teachers selected these three may have affected the results. The conclusions feel more like a summary than clear results. Maybe it is because there are no clear questions that are answered? What is the purpose of the study? This also needs to be something related to the fact that there is only ONE school being studied and that only three students were selected (or their friends were selected as well, then there were more students... as I said, it's quite messy. Furthermore, there are many additional things, but as these are on a different level and depend on what you choose to do for changes in a first processing, I will leave them be for now. I hope you have the time and opportunity to work through the article because it is an interesting question that I think more people than I would enjoy reading and being able to relate their research to, and I hope to read and comment on a new version shortly.
English is not my first language but I can easy understand the text. It is just "Bad students" that not sounds well to me (se above)
Round 2
Reviewer 2 Report
The suggestions I made in my first report have been correctly addressed.
I think the paper has improved since its first version.
I have no additional suggestions.
However, I believe that the text should be reviewed by a native translator, to avoid expressions resulting from linguistic calques of Spanish.
Best regards.
I believe that the text should be reviewed by a native translator, to avoid expressions resulting from linguistic calques of Spanish.
Author Response
Dear reviewer
We appreciate the time and the meticulous and cautious reading you have done about our document.
We are sure your comments will improve the current version because of your experience and knowledge about the topic and due to that we wanted to answer the comments that were made in your review.
Reviewer 2
Reviewers' suggestions/comments
The suggestions I made in my first report have been correctly addressed.
I think the paper has improved since its first version.
I have no additional suggestions.
However, I believe that the text should be reviewed by a native translator, to avoid expressions resulting from linguistic calques of Spanish.
Response1 (changes done/ justification):
The document is attached, revised and adapted by a native-speaking translator.
Warm regards.
The authors.

Reviewer 3 Report
Dear authors!
Thanks for letting me read your updated script. I think overall the text is much clearer now. Table two makes the method much clearer and you have written more clarifications about the representations of "bad students". The fact that you highlight the limitations at the end also makes the implications clearer. However, I am still concerned about the quotes and references. You write that you added several English references, but I can only find one in the reference list. As I mentioned in the first review, I see a great advantage in you choosing to write and publish in English to also reach the "non-Spanish speaking part of the world". But as a researcher whose knowledge of Spanish extends to having interrogated my children on Spanish glosses a few times, an article that almost exclusively has Spanish references is something I generally do not read very carefully because it becomes difficult for me to determine its "validity" in relation to other research. Here I recognize names such as Jodelet and Moscovici that I read in English, Moscovisi's thesis from 1976 is, for example, translated. Well, maybe this is not really important for this publication but more of an advice for the future. Regarding the quotes, this is perhaps also a question to discuss with the editor of the journal whether you should only cite English sources (and rewrite the Spanish ones in your own English words without citing), if you keep current quotes but put a footnote that it is your own translation or if it is ok to leave the quotes as they are. In summary, a well-executed processing in which I think you have succeeded in elevating the text according to my wishes without renouncing your own contribution. Well done!
Author Response
Dear reviewer
We appreciate the time and the meticulous and cautious reading you have done about our document.
We are sure your comments will improve the current version because of your experience and knowledge about the topic and due to that we wanted to answer the comments that were made in your review.
Reviewer 3
Reviewers' suggestions/comments
- Thanks for letting me read your updated script. I think overall the text is much clearer now. Table two makes the method much clearer and you have written more clarifications about the representations of "bad students". The fact that you highlight the limitations at the end also makes the implications clearer. However, I am still concerned about the quotes and references. You write that you added several English references, but I can only find one in the reference list. As I mentioned in the first review, I see a great advantage in you choosing to write and publish in English to also reach the "non-Spanish speaking part of the world". But as a researcher whose knowledge of Spanish extends to having interrogated my children on Spanish glosses a few times, an article that almost exclusively has Spanish references is something I generally do not read very carefully because it becomes difficult for me to determine its "validity" in relation to other research. Here I recognize names such as Jodelet and Moscovici that I read in English, Moscovisi's thesis from 1976 is, for example, translated. Well, maybe this is not really important for this publication but more of an advice for the future. Regarding the quotes, this is perhaps also a question to discuss with the editor of the journal whether you should only cite English sources (and rewrite the Spanish ones in your own English words without citing), if you keep current quotes but put a footnote that it is your own translation or if it is ok to leave the quotes as they are. In summary, a well-executed processing in which I think you have succeeded in elevating the text according to my wishes without renouncing your own contribution. Well done!
Response 1 (changes done/ justification):
References were included in English (2) and French (2).
Proofreading by a native translator has been included.
An explanatory note has been included at the end of the text: the textual quotations from the references have been translated from Spanish by the authors.
Warm regards.
The authors.
